



# Satellites reveal a 28% drop in Ukraine's Nitrogen oxides emissions during the Russia-Ukraine war in 2022

Yu Mao[1], Weimin Ju[1,4], Hengmao Wang[1], Liangyun Liu[2], Haikun Wang[3], Shuzhuang Feng[1], Mengwei Jia[1], Fei Jiang[1,4,5*]

[1] Jiangsu Provincial Key Laboratory of Geographic Information Science and Technology, International Institute for Earth System Science, Nanjing University, Nanjing, 210023, China
[2] Aerospace Information Research Institute, Chinese Academy of Sciences, Beijing, 10094, China
[3] School of Atmospheric Sciences, Nanjing University, Nanjing, 210023, China
[4] Jiangsu Center for Collaborative Innovation in Geographical Information Resource Development and Application, Nanjing, 210023, China
[5] Frontiers Science Center for Critical Earth Material Cycling, Nanjing University, Nanjing, 210023, China

*Correspondence to*: Fei Jiang (jiangf@nju.edu.cn)

**Abstract.** The outbreak of the Russia–Ukraine war in 2022 brought a huge impact on the Ukrainian economic production. To quantify this effect, we invert the anthropogenic Nitrogen oxides ($NO_x$) emissions in Ukraine from 2019 to 2022, a key indicator of human activities, to reflect the disruption of activities in different economic sectors due to war. We found a 28% decline in $NO_x$ emissions during the war, if compared with the base year, which significantly exceeded the decrease caused by the 2020 COVID-19 pandemic. Eastern Ukraine experienced a 34% decrease in $NO_x$ emissions, whereas the other regions experienced a decrease of 24%. The destruction of infrastructure and energy shortages severely impact the sustainable development of such social activities as industry, housing and transportation in Ukraine. These findings highlight the severe disruption of socio-economic activities due to the war, offering crucial insights into the broader implications of war on environmental and economic stability.

## 1 Introduction

In recent years, localized conflicts have proliferated, posing persistent challenges to economic stability, infrastructure, and social sustainability (Esteban et al., 2012; Gutierrez et al., 2024; Hou et al., 2024) . Accurately and swiftly assessing the immediate and long-term impacts of modern warfare on human society has become increasingly critical. The ongoing Russia–Ukraine war, one of Europe's most significant conflicts since World War II (Adekoya et al., 2022), provides a unique lens for studying the consequences of contemporary warfare.

Since its outbreak on February 24, 2022, this conflict has profoundly disrupted Ukraine's economy, environment, and societal well-being (Ialongo et al., 2023; Kussul et al., 2023; Malarvizhi et al., 2023; Shumilova et al., 2023). Preliminary assessments of the damage caused by the conflict have already been made by a number of organizations. As of early 2023, the Office of the United Nations High Commissioner for Human Rights (OHCHR) reported nearly 22,000 civilian casualties



(OHCHR, 2023). The war has significantly impacted all sectors, with the destruction of infrastructure, residential areas, and industrial facilities particularly severe. The resulting disruptions have not only affected Ukraine but also had global repercussions, including food shortages and energy export restrictions, which have in turn altered global energy and food

systems (Carriquiry et al., 2022; Chen et al., 2023; Huang et al., 2023; Mottaleb et al., 2022; Rawtani et al., 2022). These events have also contributed to the expansion of global cropland and the loss of biodiversity (Chai et al., 2024). However, the accuracy and completeness of the available assessment data remain uncertain. This uncertainty stems from significant challenges in acquiring reliable and up-to-date statistics due to the deteriorating information environment during the conflict, making it difficult to assess the spatial and temporal dynamics of the war's impacts (Dando et al., 2023; Hou et al., 2024;

Mueller et al., 2021). Consequently, there is an urgent need for remote, real-time quantitative methods to assess the extent of destruction across various regions at different stages of the conflict. Such approaches are crucial for supporting the well-being of civilians affected by the war.

Nitrogen oxides ($NO_x = NO + NO_2$) are significant air pollutants that reflect shifts in energy consumption, thereby serving as indicators of economic growth (Bilgen, 2014). Anthropogenic $NO_x$ emissions primarily arise from fossil fuel combustion

activities (Miyazaki et al., 2012; Zhang et al., 2023). Emissions sources include energy consumption in residential settings (e.g., natural gas) (Lebel et al., 2022), industrial production (Gholami et al., 2020; Li et al., 2023b; Zhu et al., 2023), energy supply (e.g., power plants) (Islam et al., 2023; Liu et al., 2015; Tang et al., 2019), and agriculture (Chen et al., 2018; Liu et al., 2018). With increasing motorization and urbanization, the transportation has emerged as the largest contributor to $NO_x$ emissions (Wu et al., 2017), accounting for approximately 42% of total $NO_x$ emissions in Europe (Sun et al., 2018).

Consequently, $NO_x$ emissions often represent changes in the intensity of activities in various economic sectors. As a short-lived gas, directly emitted nitric oxide (NO) rapidly oxidizes to form nitrogen dioxide ($NO_2$), which, in the presence of sunlight and oxidized volatile organic compounds (VOCs), contributes to net ozone ($O_3$) generation (Chameides, 1978; Crutzen, 1970). This short-lived nature typically results in a strong correlation between atmospheric $NO_2$ concentrations and $NO_x$ emissions in localized regions (Richter et al., 2005). This relationship provides a theoretical basis for the grid-scale

inversion of $NO_x$ emissions from satellite-observed atmospheric $NO_2$ data (Reuter et al., 2014).

The characteristics of satellite data, including their large-scale coverage and timeliness, ensure effective data support even under unforeseen circumstances, providing valuable insights into the spatial pattern and scale of global $NO_x$ emissions through top-down inversion (Li et al., 2023a). Various inversion techniques, including plume, Gaussian, and box models, permit the estimation of surface $NO_x$ emissions directly from satellite-derived $NO_2$ vertical column density (VCD) (Beirle et

al., 2011; Benjamin de Foy et al., 2014; Duncan et al., 2013). Moreover, approaches that integrate satellite observations with atmospheric chemistry transport models, such as mass balance methods, 4D-Var methods, and ensemble Kalman filtering methods, facilitate a more accurate characterization of spatial and seasonal emission trends (Gu et al., 2016; Martin et al., 2003; Miyazaki et al., 2012; Qu et al., 2017; Stavrakou et al., 2013; Xu et al., 2013). Studies have revealed the impact of the COVID-19 pandemic on human activity and the spatial and temporal dynamics of regional economies based on inverted $NO_x$





emissions from satellite observations (Feng et al., 2020; Mao et al., 2024; Miyazaki et al., 2021; Zheng et al., 2021; Guevara et al., 2021; Li et al., 2023a; Li and Zheng, 2023; Luo et al., 2023).  Analyzing changes in $NO_x$ emissions in Ukraine during the war period facilitates a quantitative assessment of disruptions in industrial output, transportation activities, residential energy consumption, and broader implications for sustainable development.

In this study, we optimise the inversion framework we previously published (Mao et al., 2024) to invert changes in

anthropogenic $NO_x$ emissions during the 2022 war in Ukraine using observations of $NO_2$ VCD from the TROPOspheric Monitoring Instrument (TROPOMI) satellite (Van Geffen et al., 2022). The framework utilizes the Community Emissions Data System (CEDS) anthropogenic $NO_x$ emission inventory(Hoesly et al., 2018) as the prior emission and simulates the atmospheric $NO_2$ transport process using the GEOS-Chem atmospheric chemistry model (GEOS-Chem 14.0.0, 2022). We employ the finite-difference mass-balance (FDMB) method (East et al., 2022) to establish a linear relationship between

anthropogenic $NO_x$ emissions and satellite-derived $NO_2$ column concentration observations, enabling us to invert anthropogenic $NO_x$ emissions during the war in Ukraine (see Materials and methods). Our analysis captures the spatial and temporal variability of $NO_x$ emissions across the industrial, agricultural, transportation, and residential sectors in Ukraine from 2019 to 2022. This approach allows us to assess the changing intensity of the war's impact on various economic activities within Ukraine. For comparison, we also analyzed $NO_x$ emissions in 2020, which were affected by the pandemic.

In addition, we divided Ukraine into eastern and western regions based on the scope of the war and estimated the $NO_x$ emissions from different sectors in each region. By conducting a comparative analysis of $NO_x$ emission changes across regions and sectors during the war, we identified the driving factors behind these changes, revealing the profound economic disruptions caused by the war. This analysis underscores the importance of sustainable development strategies and resilience planning in mitigating the impacts of such crises on modern society.

**2. Materials and methods**

**2.1 Atmospheric Chemical Transport Model**

We utilized version 14.0.0 of the GEOS-Chem model (Melissa, 2022), to conduct forward simulations of $NO_2$ VCDs in the troposphere over Ukraine, which has been widely used in the inversions of surface $CO_2$ flux (Wang et al., 2022), $CH_4$ (Shen et al., 2023)  and $NO_x$ emissions (Zheng et al., 2020). Specifically, local simulations for the European region were conducted

within the geographical bounds of 30°W–50°E longitude and 30°N–70°N latitude. The boundary conditions were derived from global simulations conducted using the same version of the GEOS-Chem model. The model was operated at a horizontal resolution of 0.5° × 0.625° and a vertical resolution of 47 layers. Meteorological data from the Modern-Era Retrospective Analysis for Research and Applications, version 2 (MERRA-2) was employed to drive the model, while the natural source $NO_x$ emissions were based on the model's default emission inventory.





## 2.2 Prior Emissions

The anthropogenic $NO_x$ emissions inventory used in this study originated from the Community Emissions Data System (CEDS) developed by the Pacific Northwest National Laboratory and the University of Maryland's Joint Institute for Global Change Research (Hoesly et al., 2018). In the inversion process, the inventory was repurposed by reallocating emissions across sectors. Specifically, emissions from industrial sources and sporadic energy consumption were combined into a single category. Emissions from the residential sector were derived from the residential, commercial, and waste disposal sources outlined in the prior inventory. The agricultural and transport sectors remained consistent with the initial inventory. It is also pertinent to note that emissions from ships and aviation were excluded from the prior inventory, which aligns with this study's focus on anthropogenic emissions originating from land sources.

In this study, the biomass burning $NO_x$ emissions data was derived from the GFED4 inventory, which is based on the Global Fire Emissions Database (GFED)(Randerson et al., 2018). GFED4 provides global data on monthly burned area at a 0.25° spatial resolution, using a combination of MODIS burned area maps, active fire data from the Tropical Rainfall Measuring Mission (TRMM), and the Along-Track Scanning Radiometer (ATSR) sensor family(Giglio et al., 2013). This inventory offers significant improvements over previous versions by incorporating a higher spatial resolution and more accurate fire mapping, which is critical for estimating emissions from biomass burning. The GFED4 inventory accounts for the dynamics of biomass burning and the associated emissions, including $NO_x$, by mapping burned areas at fine spatial and temporal scales. This allows for the assessment of interannual variability and long-term trends in biomass burning. The inclusion of biomass burning emissions is essential for understanding the contributions of wildfires and other biomass combustion sources to global $NO_x$ levels, especially in regions affected by seasonal fires.

The soil $NO_x$ emissions used in this study are based on a modified version of the Berkeley-Dalhousie Soil Nitrogen Oxide Parameterization (BDSNP), originally developed by Hudman et al. (2012) and implemented in GEOS-Chem. This emission inventory represents a significant advancement over previous parameterizations by adopting a more mechanistic approach to modeling soil $NO_x$ emissions. The BDSNP accounts for the persistent dependence of emissions on soil moisture and temperature, as well as the pulsed emissions following soil wetting events, which are critical for capturing the temporal variability of soil $NO_x$ emissions. The inventory also includes detailed spatiotemporal representations of nitrogen inputs from fertilizers, manure, and atmospheric deposition.

## 2.3 Satellite NO₂ VCD Observations

Currently, the TROPOMI satellite provides high-quality $NO_2$ VCD data with enhanced spatial resolution and signal-to-noise ratio, which are beneficial for $NO_x$ emission studies (Sekiya et al., 2022; Veefkind et al., 2012). In this study, we employed TROPOMI v2.3.1 data to provide observational constraints for our inversion framework. TROPOMI, launched in October 2017 onboard the European Space Agency (ESA) S5P spacecraft, provides high-quality global daily observations that exhibit a strong correlation with ground-based data (Ialongo et al., 2020) and demonstrate relative stability in statistical





uncertainty (Van Geffen et al., 2020). We meticulously screened grids with daily observations of $NO_2$ VCDs in the troposphere using TROPOMI. This process ensured that the data quality threshold exceeded 0.75 and the cloud cover was below 30%. TROPOMI data were gridded using the HARP toolkit of the Coordinated Toolkit for Scientific Earth Observation Data (CTSOD), which facilitated the amalgamation of daily global observations and their alignment to the same spatial resolution as the simulated concentrations. We synchronized the concentration values for each grid of the simulated concentrations with the moment of TROPOMI transit to ensure spatial and temporal coherence between the two datasets.

Furthermore, when using TROPOMI $NO_2$ VCD as observational constraints for inversion, the quality of satellite observations directly impacts the inversion accuracy. We summarized the number of valid grids and their proportion relative to the total grid count for daily TROPOMI observations over Ukraine (Fig. S1). The results show a significant reduction in valid observations during winter, with the proportion falling below 40%. To address this, we averaged the satellite data to the monthly scale, which improved the valid grid proportion to over 90% throughout the study period (Fig. S1). Therefore, we conducted the inversion of anthropogenic $NO_x$ emissions on a monthly scale to minimize the impact of missing satellite observations.

## 2.4 Inversion method

In the previous study, we developed a two-step inversion framework for estimating g lobal anthropogenic $NO_x$ emissions (Mao et al., 2024). In this study, we optimized this framework by incorporating the treatment of natural source $NO_x$ emissions and used it to invert monthly anthropogenic $NO_x$ emissions in Ukraine from 2019 to 2022. This inversion framework is based on the FDMB method and draws on the framework developed by Zheng et al. (2020), which optimizes the prior anthropogenic $NO_x$ emissions by fitting satellite $NO_2$ VCD observations and interannual variability to simulated $NO_2$ concentrations.

The FDMB method uses a prior $NO_x$ emission inventory and simulates two scenarios using an atmospheric chemical transport model: a baseline simulation and a reduced-emission simulation, where emissions are scaled by a certain factor. By comparing the $NO_2$ VCD differences between these two simulations, we derive a scaling factor that relates $NO_x$ emissions to $NO_2$ VCDs, establishing a local correspondence between the two. This allows the conversion of the $NO_2$ VCD bias between satellite observations and prior simulations into an error estimate for the prior emissions.

In this study, for the 2019 simulation, we used the 2019 "bottom-up" CEDS $NO_x$ emission inventory as the prior inventory and simulated both the baseline (1×) and 40% reduced $NO_2$ VCDs using the GEOS-Chem CTM. From this, we estimated the emission scaling factor $\beta_{2019}$ for each month of 2019 (Eq. 1):

$$\beta_{2019} = 0.4 \div \frac{\Delta\Omega}{\Omega_{sens}} \tag{1}$$




Where $\Delta\Omega = \Omega_{base} - \Omega_{sens}$, with $\Omega_{base}$ being the baseline simulated concentration and $\Omega_{sens}$ the scaled simulation concentration. Using the scaling factor $\beta_{2019}$, we constrained the 2019 prior emission $E_{prior\_2019}$ by TROPOMI observations to simulate NO₂ concentrations and inverted to obtain the posterior NO$_x$ emissions for 2019, $E_{post\_2019}$ (Eq. 2):

$$E_{post\_2019} = \left(1 + \beta_{2019}\Delta\Omega_{bias\_2019}\right)E_{prior\_2019} \tag{2}$$

Where $\Delta\Omega_{bias\_2019} = \frac{\Omega_{sate\_2019} - \Omega_{base}}{\Omega_{base}}$ represents the relative bias between the prior simulated concentrations and satellite observations. Using the FDMB method, we optimized the 2019 anthropogenic NO$_x$ emissions inventory. However, since CEDS only updates the inventory up to 2019, the interannual variability in anthropogenic NO$_x$ emissions is larger than the uncertainty in the prior inventory. Therefore, for emissions beyond 2019, we adopted a two-step inversion method to extend the emission inventory.

First, we used the 2019 CEDS emission inventory in the GEOS-Chem CTM to simulate NO₂ concentrations from 2020 to 2022 ($\Omega_{simu,202x}$), where the meteorological conditions and natural source emissions data were synchronized with the simulation period. This indicates that the changes in NO₂ concentrations after 2019 are primarily driven by non-anthropogenic factors (Zheng et al., 2020). Since the variations in satellite-observed NO₂ VCD represent the total changes, we calculated the interannual changes in NO₂ VCD caused by anthropogenic NO$_x$ emissions ($\Delta\Omega_{annu}$) (Eq. 3), and converted

these changes into interannual variations in anthropogenic NO$_x$ emissions using β_2019. This allowed us to extend the 2019 prior inventory to 2022 ($E_{prior\_202x}$) (Eq. 4).

$$\Delta\Omega_{annu} = \frac{\Omega_{sate,202x}}{\Omega_{sate,2019}} - \frac{\Omega_{simu,202x}}{\Omega_{simu,2019}} \tag{3}$$

$$E_{prior\_202x} = \left(1 + \beta_{2019}\Delta\Omega_{annu}\right)E_{prior\_2019} \tag{4}$$

Where $\Omega_{sate,202x}$ and $\Omega_{sate,2019}$ are the satellite-observed NO₂ VCDs for the corresponding months in 2020~2022 and 2019.

The inherent uncertainties in the prior inventory were not optimized and may propagate into the emission inventory for 2020~2022. As a result, we repeated the inversion process for 2019 and simulated two scenarios for the 2020~2022 NO₂ VCDs. Using the FDMB method, we inverted to obtain the posterior anthropogenic NO$_x$ emissions for each of the years.

In this study, a key advancement of our framework is the explicit decoupling of natural and anthropogenic sources. Unlike conventional approaches that fix natural emissions as static inputs, we simulate scenarios with a 40% curtailment of

emissions with the same percentage of curtailment of natural source emissions and apply the same inversion constraints to both source categories during inversion, ensuring unbiased attribution of observed NO₂ changes. Subsequently, its prior inventory scaling was used to differentiate NO$_x$ emissions from natural sources from anthropogenic sources, effectively reducing crosstalk between natural variability and anthropogenic signals.

In addition, we evaluated the changes in emissions from different sectors. Given the absence of data characterizing the

spatial and temporal changes in emissions in each sector from 2020 to 2022, we estimated the sectoral allocation. In accordance with the methodology proposed by Zheng et al. (2020), the initial allocation of optimized emissions was based on the proportion of total emissions attributed to each sector in the previous emission inventory. The dominant sector in each



grid was assumed to remain constant over time. However, of note, there is a possibility of spatial shifts in sectoral emissions over time. Therefore, we identified grids in Ukraine where sectors dominated, computed the emission changes in these regions relative to the prior inventory, and derived a relative change factor to adjust the allocation to each sector. This method accounts for both the spatial distribution of emissions by sector and temporal changes and offers a valuable approach for estimating emissions from different sectors in the absence of comprehensive data.

## 2.5 Uncertainty

To quantify the prior uncertainty remaining in the inversion system, we conducted an observing system simulation experiment (OSSE) (Atlas, 1997). In the OSSE, we used the CEDS emissions in 2019 as the prior, assuming that the true emissions in 2019 were 1.2 times the prior, and the true emissions in 2022 were 0.7 times the prior. The pseudo-true values for 2019 and 2022 were simulated using the GEOS-Chem model and processed into the same grid distribution as the TROPOMI observations described in Section 2.3, serving as pseudo-observations. These pseudo-observations were then used to constrain the prior emissions in the inversion framework, allowing us to quantify the reduction in prior bias achievable by the inversion system. For comparison, we also applied the same pseudo-observations to constrain the inversion framework from our previous study (Mao et al., 2024) to evaluate the impact of including natural source emissions on the accuracy of anthropogenic $NO_x$ emissions. Another major source of uncertainty in our results is the uncertainty in the satellite observations. The specified random uncertainty for individual TROPOMI tropospheric $NO_2$ VCD measurements is between 25% and 50%, with a precision of $0.7 \times 10^{15}$ molec cm$^{-2}$ (Malytska et al., 2024). And TROPOMI $NO_2$ observations provide precision estimates for each grid.

In our results, we converted the precision range of $NO_2$ VCD observations for each grid into a corresponding precision range for the anthropogenic $NO_x$ emission constraints using the conversion factor β. We then summed the remaining prior uncertainty for each grid, as computed in the OSSE, to determine the uncertainty range in the inversion results.

## 2.6 Evaluation of the Inverted Anthropogenic $NO_x$ Emissions

Validating the accuracy of the inverted anthropogenic $NO_x$ emissions is challenging due to the limited availability of independent anthropogenic $NO_x$ emission inventories and surface observations specifically for Ukraine. To assess the robustness and reliability of the inversion results, we used a multi-pronged approach. First, we performed simulations of both prior and posterior tropospheric $NO_2$ VCDs using the GEOS-Chem model, which accounts for atmospheric chemistry and transport processes. These simulated concentrations were then compared to the TROPOMI satellite-observed $NO_2$ VCDs. This comparison allowed us to assess the temporal and spatial consistency of the inversion framework against high-resolution satellite observations.

Additionally, as the scope of our inversion extended over the European land area, we leveraged independent in-situ observations of surface $NO_2$ concentrations from the European Environment Agency (EEA). These ground-based



observations served as a critical reference to validate the simulated surface concentrations corresponding to the posterior

$NO_x$ emissions.   Specifically, we fitted the simulated surface $NO_2$ concentrations to the time series of in-situ measurements for multiple stations across Europe, facilitating an evaluation of how well the prior and posterior simulations captured the observed $NO_2$ variability. To quantify the accuracy of these fits, we performed a linear regression analysis comparing the time-averaged surface concentrations from the modeled grid cells with the in-situ measurements. We then evaluated the fitting accuracy by examining the $R^2$ values and root mean square errors (RMSE) between the simulated and observed

concentrations.

Moreover, to assess the temporal stability and robustness of the inversion framework, we compared the interannual variations in the relative deviations between the posterior results and both satellite and ground-based observations. By examining how the relative biases evolved over time, we were able to ensure that the posterior emissions remained consistent with observational trends.

**3. Results**

**3.1 National Decline**

To ascertain the impact of the Russia–Ukraine war on anthropogenic $NO_x$ emissions in Ukraine, we compared the monthly variations in anthropogenic $NO_x$ emissions during the war with pre-war levels (baseline emissions). We derived baseline emissions by averaging the emissions in 2019 and 2021, excluding 2020 owing to the influence of the COVID-19 pandemic.

Fig. 1 depicts the seasonal variations in $NO_x$ emissions in 2022 and the changes relative to baseline years. In 2022, Ukraine exhibited a 24% (±7.6%) reduction in $NO_x$ emissions compared with the baseline, and the reduction during the war period (March to December) was 28% (±7.0%). The first sharp decline of 41% (±16%) occurred in March, coinciding with the start of the full-scale Russian invasion in late February 2022, indicating a significant short-term societal disruption. Subsequently, the rate of decline in emissions slowed, stabilizing at approximately 15% (±3.8%) from June to October. This stabilization

was due to the stalemate of the war in eastern regions of Ukraine, which suggests a rapid societal adaptation within a month of the war's outbreak, with partial restoration of social functions and preparations for long-term war. After October, the $NO_x$ emissions showed a new round of decline, reaching a peak decline of 55% (±24%) in December. This was primarily due to the increased energy demand in the baseline years and intensified energy shortages in 2022 during the cold season. The war-induced constraints on energy supply and population migration caused the seasonal emission pattern in 2022 to differ

significantly from that of the baseline (Fig. 1a).





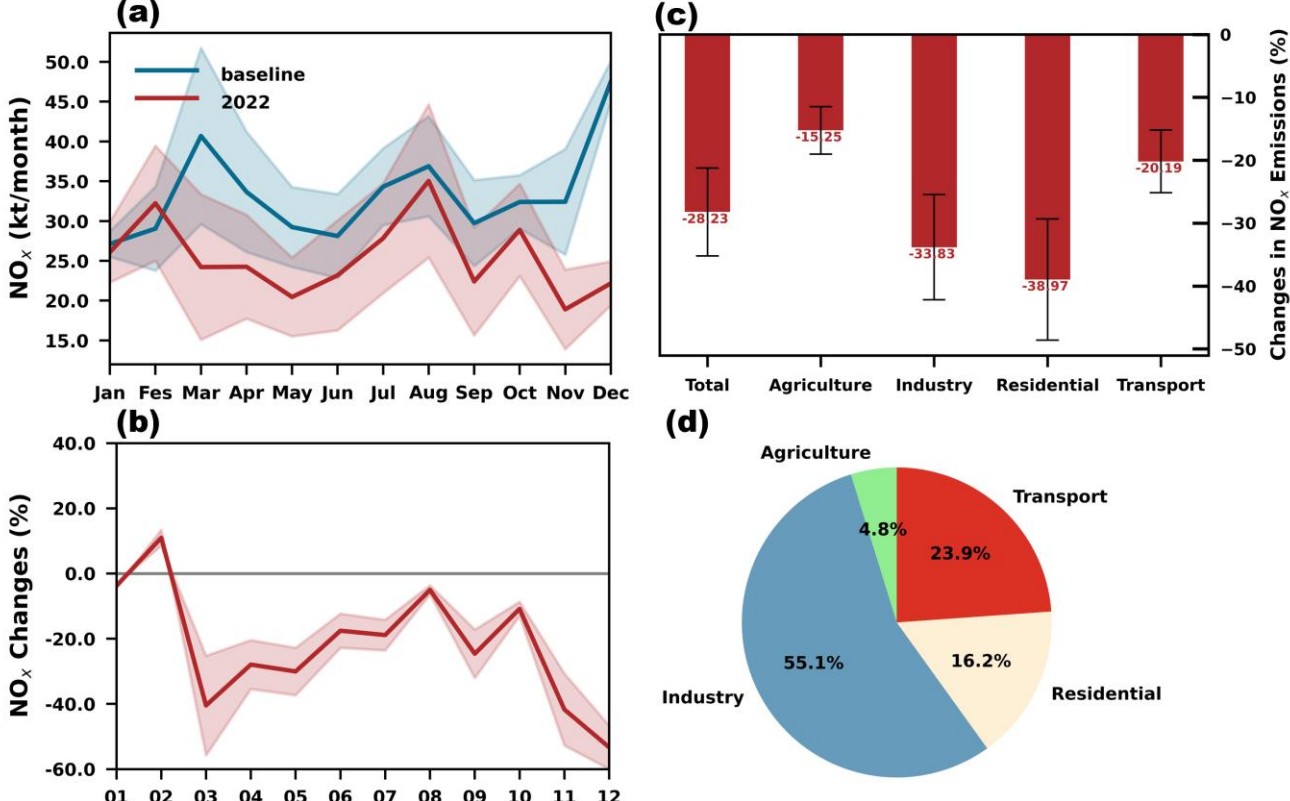

**Fig. 1. Changes in total NO$_x$ emissions and in different sectors throughout Ukraine in 2022.** (a) Monthly NO$_x$ emissions in 2022 compared to the baseline. (b) Relative change in monthly emissions in 2022 compared to the baseline. (c) Relative change in emissions from different sectors during war compared to the baseline. (d) Contribution of different sectors to the decrease in emissions.

We examined the reduction in sectoral NO$_x$ emissions during the war relative to the baseline (Fig. 1c) and evaluated each sector's contribution to the overall emissions reduction (Fig. 1d). This war resulted in a reduction in emissions across all the sectors of Ukraine. The industrial sector experienced the most significant impact, accounting for 55% of the total decline, decreasing by 34% (±8.4%) compared with the baseline. This decline was primarily due to the impact of the ongoing war in eastern regions, where industry is well developed. The reduction in residential emissions was also notable, with a 39%

(±9.6%) decline compared to the baseline, largely attributable to depopulation during the war. The transport sector was a significant contributor to land-based anthropogenic NO$_x$ emissions in Ukraine, but the observed decline was not as pronounced as that observed in the residential and industrial sectors. The reduction in transport emissions was partially offset by emissions resulting from population movement and resource transfers. Moreover, the war-affected eastern and southern Ukraine, which are the dominant wheat producers (Carriquiry et al., 2022), exhibited a considerable decline in agricultural

emissions.



We compared the seasonal variations in sectoral emissions between the war in 2022 and the baseline (Fig. S2). Emissions across all sectors exhibited a comparatively smaller decline during the summer months. Agricultural emissions were significantly affected in the initial stages of the war (March–April) and during the cold season (November–December), with decreases of 26% (±9.6%) and 52% (±15.8%), respectively. By contrast, the war-induced impact on $NO_x$ emissions was less

pronounced during the warm season (1.4%±3.8%). The industrial emissions exhibited a notable decline of 39% (±19.6%) in March, followed by a relatively stable period. A significant 70% (±14.9%) decline was observed in emissions from the residential sector during the first three months of the war. Furthermore, a further decline in emissions occurred during the winter months owing to energy shortages. The transportation sector demonstrated notable responsiveness to increased transportation demand during the war, with a 28% (±16.2%) decline in emissions observed in March, followed by a slight

decrease in April. Nevertheless, the decline in emissions remained at approximately 14% (±7.5%), owing to the destruction of infrastructure and a shortage of drivers.

## 3.2 Decline in Different Regions

We examined the spatial distribution of total emissions (Fig. 2a) and monthly $NO_x$ emissions (Fig. S3) during the war relative to the baseline. Most regions of Ukraine exhibited a reduction in $NO_x$ emissions of approximately 30% (Fig. 2a).

This reduction was more pronounced in eastern regions of Ukraine than in the central and western regions. A slight increase in emission was observed in parts of the Kyiv, Zhytomyr, Rivne, and Khmelnytskyi oblasts (Fig. 2a). Increases in the $NO_2$ VCDs in these regions were observed using the TROPOMI satellite (Fig. S4), which were mainly from the agricultural and transportation sectors (Fig. S5), possibly because these regions are located away from the front of the war around the Ukrainian capital, where a larger population has gathered. After April 2022, the primary theater of the Russia–Ukraine war

was concentrated in the eastern oblasts bordering Russia. The central and western regions exhibited a weaker impact than the eastern regions. In terms of the spatial distribution of sectoral variations (Fig. S5), the most pronounced decline in agricultural emissions was observed in eastern and southern Ukraine. This may be attributed to the region's importance as a winter wheat-producing area (Lin et al., 2023) and the disruption of cultivation caused by the Russian army's control, which caused a reduction in fertilizer application and agricultural $NO_x$ emissions. Spatially, industrial emissions exhibited a notable

decline throughout Ukraine, particularly in Luhansk Oblast. The war had a direct or indirect impact on industrial production across Ukraine, with the most pronounced damage observed in conflict zones. The outbreak of the war during this period resulted in the destruction of numerous Ukrainian residential facilities and the displacement of numerous refugees to the central and western regions and abroad. Consequently, emissions increased as the war stabilized in the eastern regions, housing conditions improved, and some refugees returned home.

To assess the geographical difference in $NO_x$ emissions during the war in Ukraine, we categorized Ukraine into Eastern Ukraine and Western Ukraine based on the scope of the war and analyzed the seasonal variations in satellite $NO_2$ VCDs in both regions (Fig. 2a). Subsequently, we calculated the seasonal fluctuations in anthropogenic $NO_x$ emissions in comparison





with the baseline for different regions (Fig. 2 b–e). Both the Eastern Ukraine and Western Ukraine emissions exhibited a notable decline in March, with reductions of 40% (±15%) and 41% (±23.7%), respectively, caused by a large-scale attack
that began in late February 2022. Subsequently, Western Ukraine emissions exhibited a gradual recovery from April to August, whereas Eastern Ukraine emissions remained low in the subsequent months, reaching their lowest (46%±11%) in May. This discrepancy may be attributed to the initial direct impact of the war on the Western Ukraine in March, which resulted in a rapid decline in emissions. The Eastern Ukraine, with its smaller population and dominant industrial sector, did not minimize emissions in the first month. However, as the war continued, emissions continued to decline due to population
loss, energy shortages, and infrastructure damage. In both regions, the decline in emissions intensified concurrently after November, likely because of the inability of energy shortages to meet increased demand during the winter months.

A comparison of sectoral emissions in the Eastern Ukraine and Western Ukraine (Fig. S6) revealed that the industrial and residential sectors were the most significantly impacted emission sectors in both regions. This underscores the emergence of energy shortages as a major challenge for Ukraine amid the ongoing war. Due to the geographical ramifications of the war,
all sectors in Eastern Ukraine experienced more pronounced declines in emissions than Western Ukraine. In Western Ukraine, except for the industrial and residential sectors, which exhibited reductions of 32% (±7.9%) and 36% (±9.0%), respectively, the other sectors exhibited reductions not exceeding 20% (±1.5%). Nevertheless, the agricultural and transport sectors contributed more to the overall decline in emissions in Western Ukraine than they did in Eastern Ukraine. This was primarily attributed to the region's more advanced agriculture and larger population.





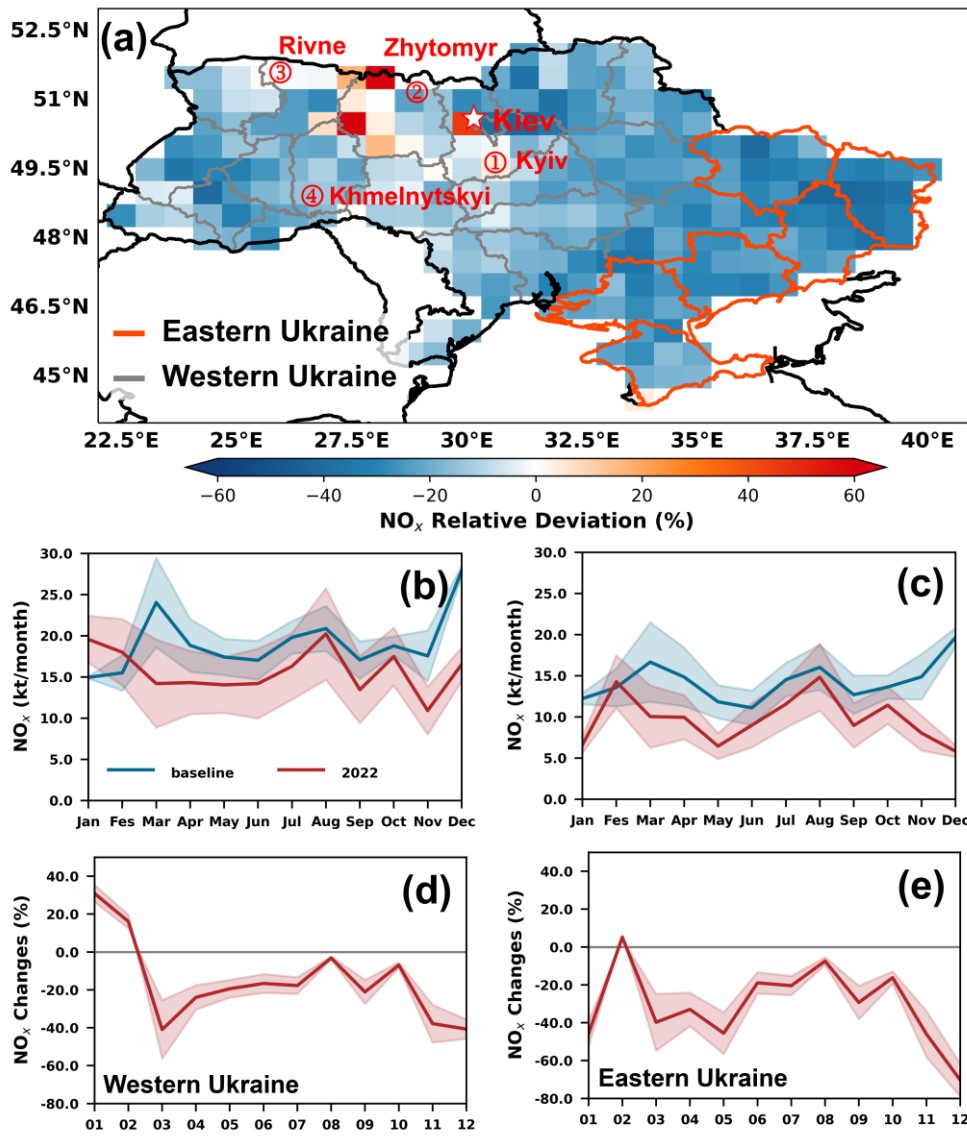

**Fig. 2. NO$_x$ emissions changes in different regions.** (a) Spatial distributions of NO$_x$ emissions changes in Ukraine during war relative to the baseline. Eastern Ukraine is marked with red lines, and Western Ukraine is in grey. Cities with increased emissions and the capital city are also marked. (b) Monthly NO$_x$ emissions in 2022 and the baseline years in Western Ukraine, (c) monthly emissions in Eastern Ukraine, (d) relative changes in monthly NO$_x$ emissions in 2022 relative to the baseline in, Western Ukraine, and (e) relative changes in Eastern Ukraine.

## 3.3 Comparison With the COVID-19 Pandemic

To provide a more comprehensive representation of the impact of the war in Ukraine, we further analyzed changes in NO$_x$ emissions during the 2020 COVID-19 pandemic. NO$_x$ emissions in Ukraine during the pandemic (April–December) in 2020 decreased by 13% (±4.1%) compared with those in the baseline year. After falling by 39% (±12.8%) for the first time in



April, NO$_x$ emissions in Ukraine in 2020 returned to levels comparable to the baseline in May, with a decline of only 3% (±0.8%) (Fig. S7). The spatial distribution of anthropogenic NO$_x$ emissions in Ukraine during the pandemic was more homogenous, with no significant east–west differences. During the pandemic, the main declining sectors were industry and transportation (15%±3.8%). The agricultural and residential sectors displayed an upward variation in most regions, whereas decreases primarily occurred in the Kiev periphery and southeast regions (Figs. S4, S8). The changes in anthropogenic NO$_x$

emissions during the pandemic are described in detail in Text S1 of the Supplementary Material.

The decline in NO$_x$ emissions owing to the war was twice as large as the impact of the pandemic. This was followed by further declines in the individual months after May, which were less pronounced than those observed in 2022. Sectoral emissions were not affected to the same extent during the pandemic as during the war. The industrial and transport sectors exhibited a general decline, primarily because of the implementation of home quarantine policies, which resulted in a

reduction in residential mobility and industrial production. This comparison highlights the more profound and enduring impact of the war on Ukraine compared to the pandemic. The pandemic slowed national human activity without causing far-reaching damage.

## 4. Discussion

Our study inverts changes in anthropogenic NO$_x$ emissions during the Russia- Ukraine war in 2022 based on satellite

observations. We explore the capacity of satellite-based NO$_x$ emission inventories to monitor economic production activities within Ukraine-affected regions amid the backdrop of frequent localized conflicts.

We conducted a comprehensive validation to ensure the reliability of the inverted NO$_x$ emissions in Ukraine. Compared with the satellite-observations, the accuracy of NO$_2$ VCDs simulated using posterior emissions significantly improved; the average discrepancies between the prior and posterior simulated NO$_2$ VCDs were 1.32 Pmolec/cm$^2$ and 0.13 Pmolec/cm$^2$,

respectively (Figs. S9, S10). The posterior emissions more accurately reflect the seasonal fluctuations in NO$_2$ VCDs in Ukraine (Fig. S9). Prior emissions exhibited an overestimation of 80.3% at the national level, with the greatest discrepancy observed in the Central and Southern regions. Furthermore, we used surface in-situ observations to assess the accuracy of the simulated NO$_2$ concentrations with prior and posterior emissions. The simulated concentrations with posterior emissions exhibited higher R$^2$ and lower RMSE values at most sites across Europe (Fig. S11 a, b). Furthermore, the simulated values of

a posterior emissions exhibited greater accuracy in fitting between sites than the prior emissions (Fig. S11 c).

Despite the improvement over the prior inventory, some discrepancies between the posterior simulations and both satellite and surface in-suit observations remain. To assess the impact of these discrepancies on the interannual variability of the results, we calculated the relative deviation between the simulated and observed values across different years, and computed the seasonal variations in the relative deviations on a monthly basis (Fig. 3). The results show that the relative deviation

between the posterior emission simulations and satellite observations remained within ±50% across these years. The largest




fluctuations in the relative deviation were observed between January and March, with deviations stabilizing for the remaining months. In contrast, when comparing with surface in-suit observations, the relative deviations were concentrated between -40% and -80%. Although these results did not exhibit the same seasonal fluctuations observed in the satellite comparison, they demonstrated lower overall variability compared to the satellite observations. Given that the relative

deviations between the posterior simulated concentrations and satellite observations remained within a consistent range across different years, we can conclude that the inversion results maintain high consistency across the years.

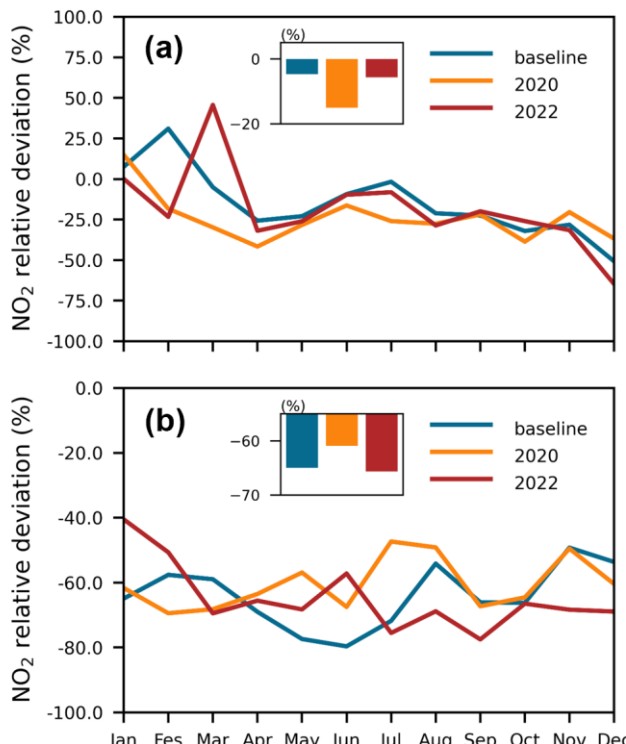

**Fig. 3. Seasonal distribution of the relative deviations in posterior simulations in the baseline, 2020 and 2022, with respect to (a) satellite observations and (b) surface in-suit observations.** The bar subplot shows the mean annual relative

deviation.

Additionally, according to the World Bank's Ukraine Rapid Damage and Needs Assessment (the World Bank, 2023), the war resulted in a 29.2% decline in Ukraine's GDP for 2022, which closely aligns with our estimated 28% reduction in emissions. We collected energy consumption data, industrial production index (IPI) data, the gross domestic product (GDP) and agricultural GDP from the State Statistics Committee of Ukraine (SSCU), and $CO_2$ emissions data from the Global Carbon

Project (GCP) (Friedlingstein et al., 2023; Jones et al., 2021) (Table 1). The SSCU and GCP provide data only for different energy types. For comparison, we assumed that oil was consumed primarily in transportation, natural gas was used primarily





by residents, and coal was mainly consumed by industry. Compared with that at baseline, the inverted $NO_x$ emissions in 2022 declined by 14%, 28%, and 32% in the transportation, residential, and industrial sectors, respectively. The relative changes in the inverted $NO_x$ emissions of each sector are highly consistent with the relative decline in oil and gas

consumption data from the SSCU. Ukraine's agricultural sector saw the smallest change in emissions in 2022, with a decline of 13%. This result is corroborated by the relatively small changes in agricultural GDP volumes reported by the SSCU. In addition, the rate of decrease in $NO_x$ emissions in the transport and residential sectors was close to the relative changes in the $CO_2$ emissions of oil and gas from the GCP, respectively. However, the $CO_2$ emissions in Ukraine in 2022 declined by 35% compared with those at baseline, which was approximately 10% more than the inverted anthropogenic $NO_x$ emissions.

According to the results in Table 1, this difference is partly due to the decline in emissions from coal combustion, which exceeds the IPI data from SSCU and the industrial sectors emissions from our inversion by about 10%. This difference may be due to the reduced utilization of coal in industrial production.

**Table 1. Changes in GCP $CO_2$ emissions and SSCU statistics for baseline, 2020 and 2022, and relative changes in 2020 and 2022 compared with the baseline.**

| Inverted $NO_x$ emissions | | | SSCU* | | | GCP $CO_2$ | | |
|---|---|---|---|---|---|---|---|---|
| Sector | Baseline | 2022 | Data type | Baseline | 2022 | Data type | Baseline | 2022 |
| Transport (kt) | 131.2 | 113.4/-14% | Oli (tb/d) | 234.5 | 200.00/-15% | Oli (Mt) | 234.5 | 200.00/-15% |
| Residential (kt) | 48.3 | 34.7/-28% | Gas (b Nm³) | 2.65 | 1.90/-28% | Gas (Mt) | 2.65 | 1.90/-28% |
| Industry (kt) | 187.5 | 128.4/-32% | IPI | 102.3 | 63.86/-38% | Coal (Mt) | 102.3 | 63.86/-38% |
| Agriculture (kt) | 34.2 | 29.6/-13% | Agricultural GDP (Mh) | 474,965 | 449,148/-5% | - | - | - |

**\* Oil data of SSCU in thousand barrels per day, Gas data of SSCU in billion Nm³, IPI calculated using 2006 as 100. Agricultural GDP in Ukraine in Millions of Ukrainian hryvnias.**

The primary sources of uncertainty in our inversion framework can be attributed to two main factors: the quality of satellite observations and the inherent limitations of the inversion process. To enhance data reliability, we utilized TROPOMI $NO_2$ VCD observations processed using a more accurate processor version. Despite this improvement, TROPOMI observations

still substantially underestimate tropospheric $NO_2$ levels (East et al., 2022). To quantify the uncertainty introduced by satellite $NO_2$ VCD observations, we estimate that these uncertainties contribute an average of ±6.5% uncertainty to the inversion results. Further, the inversion framework itself introduces uncertainty, which stems from both the missing data of satellite, particularly during the winter months at high latitudes such as in Ukraine, and the process used to derive emissions, and we quantified this part of the uncertainty using the OSSE. The OSSE results show that the inversion framework

effectively reduces the uncertainty in the prior and maintains high spatiotemporal consistency with the assumed true values (Fig. S12). The bias between the posterior emissions and the true values was 0.69%, a 98.27% reduction from the bias in the prior. Additionally, the posterior uncertainty without considering natural sources was 1.08%, reducing the prior uncertainty by 97.24%. Seasonally, by accounting for natural source emissions, the inversion uncertainty for anthropogenic NOx





emissions decreased 1.03% for the year-round, with the largest improvement observed from June to August when natural
emissions are higher, with an average improvement of 18.2% in July. However, the OSSE also revealed that despite the model's effectiveness in reducing prior error, 2.3% of uncertainty remained. Thus, after combining the uncertainties from both sources, the total uncertainty in the reduction of emissions during the war period in the inversion results is estimated to be approximately 7.6%. The uncertainties across different stages, sectors, and regions are provided in the results.

## 5. Conclusion

Our results indicate a notable reduction in $NO_x$ emissions during the war, highlighting the severe impact of the war on the Ukraine's socioeconomic activities. Sector-specific analysis revealed the most significant declines in the industrial and residential sectors. And the eastern Ukraine experiencing more severe disruptions. Additionally, the energy shortages during winter exacerbated the decline in emissions, illustrating the compounded effects of infrastructure damage and resource scarcity. The stalemated war has led to substantial declines in Ukraine's energy and industrial production levels, causing
significant damage to housing and transportation infrastructure. Rural areas and agricultural production were also significantly affected, causing further economic destabilization. This disruption in both urban and rural areas resulted in population displacement towards safer regions and even other countries, exacerbating labor shortages and further stalling economic activities. In comparison, the 2020 COVID-19 pandemic resulted in less severe and shorter-term reductions in $NO_x$ emissions, primarily affecting industry and transportation due to home quarantine policies. This reduction was temporary
and primarily driven by public health measures that aimed to protect lives while allowing for a relatively quick economic rebound post-lockdown. Conversely, the destruction during the war of industrial capacity, residential areas, and transportation infrastructure, coupled with energy shortages, has not only disrupted current activities but has also compromised future development potential.

This study emphasizes the interconnectedness of environmental sustainability and socio-economic stability. Changes in $NO_x$
emissions can only partially quantify the economic impacts of the war, and the actual socioeconomic impacts of the war are likely to be far greater than estimates derived from satellite observations. While our analysis is only a snapshot of the impacts of war, the findings have far-reaching implications for further research and policy development.

## Data availability

The TROPOMI $NO_2$ VCDs data are available at https://scihub.copernicus.eu/, the full versions of the GEOS-Chem model
and the driver data are available at https://geos-chem.readthedocs.io/en/latest/. Data of energy consumption and IPI are from the Statistical Office of Ukraine at https://www.ukrstat.gov.ua/. $CO_2$ emissions of GCP (GCP-GridFEDv2023.1) are available at https://zenodo.org/records/8386803/. The inversion of anthropogenic $NO_x$ emissions in this paper are available at https://zenodo.org/records/12540012.



**Author contribution**

Y.M. and F.J. conceptualized the study. Y.M., H.M.W., and F.J. developed the methodology. Y.M., S.Z.F., and M.W.J. conducted the investigation. Y.M. performed the visualization. W.M.J. and F.J. supervised the project. The original draft was written by Y.M., and L.Y.L., H.K.W., and W.M.J. contributed to the review.

**Competing interests**

The authors declare that they have no conflict of interest.

**Acknowledgments**

The authors gratefully acknowledge the European Space Agency (ESA) and the TROPOMI team for providing the TROPOMI $NO_2$ data used in this study. The authors also acknowledge the High-Performance Computing Center (HPCC) of Nanjing University for performing the numerical calculations in this paper on its blade cluster system.

**Financial support**

This work was supported by the National Key Research and Development Program of China (Grant No: 2023YFB3907404), the National Natural Science Foundation of China (Grant No: 42377102, 42305116), Fengyun Application Pioneering Project (Grant No: FY-APP-2022.0505), and the Research Funds for the Frontiers Science Center for Critical Earth Material Cycling, Nanjing University (Grant No: 090414380031).

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
