# Peer review of "Satellites reveal a 28% drop in Ukraine's Nitrogen oxides emissions during the Russia-Ukraine war in 2022"

_EGUsphere, 2024_

## Author Comment (AC1)

Dear Reviewer:

We would like to thank you for your valuable feedback and constructive comments on our manuscript. We have carefully considered each of the referee's comments and suggestions and have revised the manuscript accordingly. In this response, we respond to all of the comments point to point. The referee's comments are listed below in black, our response is given in blue and the modification to the manuscript is listed in red. The page and line numbers for corrections are referred to the revised manuscript; the page and line numbers of the original manuscript remain unchanged. References relevant to the response are listed at the end of this document.

**Comments:**

The manuscripts is a detailed application description of the method introduced in a previous publication Mao et al 2024 (Mao, Y., Wang, H., Jiang, F., Feng, S., Jia, M., and Ju, W.: Anthropogenic $NO_x$ emissions of China, the U.S. and Europe from 2019 to 2022 inferred from TROPOMI observations, Environ. Res. Lett., https://doi.org/10.1088/1748-9326/ad3cf9, 2024).

It shows in detail where the war in the Ukraine destroyed most of the ecomic and sozial (and human) live. The results are based on the comparison of modeled columns based on prior emissions, which are than adapted to satellite observations. The study focuses on the years 2019 (a pre-covid baseline) and the year 2022. The pandemic is included as small side note. The impact of the pandemic Ukraine's $NO_x$ emissions was negligible for compared to the Russian invasion.

**Mayor comments**

According to (https://www.temis.nl/airpollution/no2col/tropomi_no2_data_versions.php, March 2025) version 2.4 or higher of the TROPOMI $NO_2$ data is reommended. There was a major version change in July 2022, I am not sure the version 2.3.1 (page 4 line 124) is appropriate for the presented study focusing on 2022. I recommend to double check the results using latest version of TROPOMI $NO_2$ data. In this context I ask for your apologies that I did not realize in the pre-review.

Response: We sincerely thank the reviewer for your rigorous review of the satellite observation version. According to the reviewers' suggestions, we have updated the satellite data. We now use the latest version of the TROPOMI $NO_2$ data. For the time period from 1 January 2019 to 25 July 2022, we use the v2.4.0 official reprocessed product, from 26 July 2022 to 12 March 2023 we use the v2.4.0 official offline product, and from 13 March 2023 to 26 November 2023 we use the v2.5.0 official offline product. For the remaining days of 2023, the v2.6.0 official offline product is used. All data were obtained from the TEMIS website (https://www.temis.nl). As noted by the reviewer, these versions incorporate significant updates to the Level-1b data and $NO_2$ processor, and they have been adopted as the operational standard, superseding all previous versions (including v2.3.1). The revised analysis confirms the reliability of our findings while aligning with

current best practices. We have updated the manuscript accordingly.

Specifically, we have revised the description of the TROPOMI data version used in this study in Section 2.3 (see Lines 140-146):

In this study, we employed the most recent versions of the TROPOMI NO$_2$ product to provide optimal observational constraints for the inversion framework. Specifically, we used the v2.4.0 official reprocessed dataset for the period from 1 January 2019 to 25 July 2022, the v2.4.0 official offline dataset from 26 July 2022 to 12 March 2023, the v2.5.0 official offline dataset from 13 March 2023 to 26 November 2023, and the v2.6.0 official offline dataset for the remaining days of 2023. These products incorporate improved Level-1b processing and retrieval algorithms and represent the most up-to-date and consistent TROPOMI NO$_2$ datasets available. All data were obtained from the TEMIS portal (TEMIS, 2025).

Correspondingly, we have updated all the emission estimates in the Results section in the revised manuscript.

**Minor comments**

1. The war has continued for 3 too long years. How did the emissions change in the second year?

Response: Many thanks for this suggestion. We have added anthropogenic NO$_x$ emission estimates for 2023 using the new version of the TROPOMI data as a constraint in the revised manuscript.

In 2023, anthropogenic NO$_x$ emissions in Ukraine declined by 7.6% (±1.4%) relative to the baseline, a smaller reduction compared to 2022. This year was characterized by increased temporal variability and more pronounced regional divergence. The most substantial declines were observed in February, April, September, and December, corresponding to periods of intensified military activity and heightened energy shortages during the cold season. In contrast, emissions exceeded baseline levels in March and from May to July, likely due to the resumption of agricultural activities and the initiation of reconstruction efforts. Agricultural NO$_x$ emissions increased by 15% (±2.7%), marking a further rise compared to 2022. Industrial emissions continued to decline significantly, with an annual reduction of 24% (±4.4%), largely driven by ongoing disruptions in the eastern conflict zones. Residential emissions remained at depressed levels (18% ±3.5%), indicating that population displacement had yet to reverse. Transport-related emissions exhibited a modest rebound in the first eight months, followed by stabilization towards the end of the year. Regionally, the eastern part of Ukraine continued to experience the largest reductions (17%), driven by industrial stagnation, destruction of energy infrastructure, and substantial population loss. In contrast, emissions in western regions declined by only 3%, reflecting greater resilience supported by industrial relocation, adaptive agricultural practices, and international assistance. Notably, eastern Ukraine experienced the lowest emission levels in January, April, and September, while June showed a smallest reduction (12%) likely linked to

military mobilization ahead of counteroffensive operations. In the west, a significant reduction was observed only in February, associated with energy supply disruptions.

We have revised the Results section of the manuscript to incorporate these changes (see Lines 287–320):

In 2023, anthropogenic $NO_x$ emissions in Ukraine declined by 7.6% (±1.4%) relative to the baseline (Fig.2 b, c, e). The most pronounced reductions were observed in February, April, September, and December. Notably, unlike the sustained emission decline throughout 2022, 2023 exhibited intermittent increases, with higher emissions than the baseline observed in March and from May to July. From a seasonal perspective, persistent energy shortages during the cold season remained a critical constraint in early spring (before March) and late autumn (after September), contributing to continued emission reductions during these periods.

[Figure]

**Fig. R1. Changes in anthropogenic $NO_x$ emissions in Ukraine during 2022 and 2023 and their deviations from the baseline period.** (a) Daily anthropogenic $NO_x$ emissions in 2022 (red) and the baseline period (black). (b) Daily anthropogenic $NO_x$ emissions in 2023 (blue) and the baseline period (blue). (c) Relative differences in daily emissions in 2022 and 2023 compared to the corresponding days in the baseline period. (d) Sectoral contributions of prior (blue) and posterior (red) emissions in 2022 relative to the baseline. (e) Sectoral contributions of posterior emissions in 2023 (orange) relative to the baseline. (f) Contribution of each sector to the total reduction in emissions in 2022 (prior and posterior) and 2023, compared with the baseline year. (Fig.2 in revised manuscript)

We examined the reductions in anthropogenic $NO_x$ emissions across different sectors following the outbreak of war in 2022 based on both the prior and posterior estimates, as well as the sectoral changes in 2023 derived from the inversion (Fig. 2d, e), and the contribution of each sector to the total emission reduction (Fig. 2f). The inversion indicate that the industrial sector experienced the most significant impact from the war. Compared to the baseline, industrial emissions declined by 34% (±6.1%) in 2022 and by 24% (±4.4%) in 2023, accounting for 72% and 106% of the total annual reductions, respectively. These declines are comparable to those estimated from the prior inventory for 2022 and can largely be attributed to the heavy fighting and infrastructure disruption in the eastern industrial regions of Ukraine. Residential emissions also showed substantial reductions of 23% (±4.1%) in 2022 and 18% (±3.5%) in 2023. Despite contributing less to the total reduction due to their relatively smaller share of emissions, the residential sector was the only sector in 2023 that did not exhibit a notable alleviation in its reduction rate. This persistence is likely associated with the population loss. The transport sector was a significant contributor to land-based anthropogenic $NO_x$ emissions in Ukraine, but the observed decline was not as pronounced as that observed in the residential and industrial sectors. The reduction in transport emissions may have been partially offset by increased emissions from population displacement and logistical movements. Moreover, compared to the prior inventory, the inversion suggests a smaller reduction in transport emissions in 2022. This discrepancy could be due to the underestimation of military and emergency transport activities in energy-based inventories. In contrast to the prior inventory, which suggested a 3% decline in agricultural emissions in 2022, the inversion results indicate a 4% (±0.7%) increase in 2022 and a more pronounced 15% (±2.7%) increase in 2023. This discrepancy likely arises from the limitations of statistical data used in the prior inventory, which may have underestimated additional $NO_x$ emissions from traditional farming practices and irregular land management under war conditions. The inversion results also suggest that the increase in agricultural emissions may partly reflect an overestimation of emissions in the central and western regions due to the assumption of fixed sectoral emission allocation.

And in Lines 329–333:

During the summer of 2023, agricultural emissions were approximately 6% higher than in 2022, indicating a gradual economic recovery in the central and western agricultural regions of Ukraine one year after the outbreak of the war. In contrast, the industrial sector exhibited substantial fluctuations in its emission reductions in 2023, likely reflecting repeated military operations in the eastern conflict zones. Transport-related $NO_x$ emissions increased by 4.3% compared to 2022, indicating a gradual recovery in domestic mobility.

For decline in different regions (Lines 351–352):

In 2023, $NO_x$ emissions in central and western Ukraine increased compared to the baseline, while emissions in the eastern regions remained suppressed due to ongoing localized conflict (Fig. 3b).

And in Lines 354–360:

In 2023, emissions increased further across the region, reflecting Crimea's growing role as a long-term base for Russian troop deployment and logistical support. Sectoral analysis reveals varying degrees of recovery in agriculture, industry, and transportation across central and western Ukraine. In contrast, residential emissions in 2023 remained at levels comparable to those in 2022, consistent with the national trend. These findings suggest that while efforts to reestablish agricultural and industrial activity have

taken hold in the rear regions of the battlefield, residential activity has yet to recover, likely due to continued population displacement.

[Figure]

**Fig. R2. NO$_x$ emissions changes in different regions.** (a~b) Spatial distributions of NO$_x$ emissions changes in Ukraine during war in (a) 2022 and (b) 2023 relative to the baseline. Eastern Ukraine is marked with red lines, and Western Ukraine is in black. (c~d) Daily NO$_x$ emissions in 2022, 2023 and the baseline years in (c) Eastern Ukraine and (d) Western Ukraine. (e~f) Relative changes in daily NO$_x$ emissions in 2022 and 2023 relative to the baseline in, (e) Eastern Ukraine, and (f) Western Ukraine. (Fig.3 in revised manuscript)

And in Lines 379-389:

In 2023, anthropogenic NO$_x$ emissions in Ukraine also exhibited a marked spatial divergence between the eastern and western regions. The Eastern Ukraine experienced a sustained emission decline of 17.0 % (±3.1%) compared to the baseline, largely attributed to industrial inactivity under Russian control, destruction of power infrastructure, and ongoing population displacement. In contrast, emissions in the Western Ukraine declined by only 2.9 % (±0.5%), reflecting greater resilience due to the westward relocation of industry, influx of international assistance, and support from adaptive agricultural practices. Seasonal trends reveal that eastern emissions reached their lowest points in January (-35.8 %±6.4%) due to exacerbated energy shortages, in April (-16.2 %±2.9%) during intensified military offensives, and in September (-31.7 %±5.7%) following a second collapse of the power grid. The smallest reduction (11.9 %±2.1%) was observed in June, likely driven by intensified military activity during Ukraine's counteroffensive. In the west, the most substantial reduction occurred in February (10.0%±1.8%) and December (22.9%±4.1%), primarily due to gas supply disruptions by Russia and regional power grid failures. Emissions in other months remained comparable to or slightly above baseline levels.

2. page 5 line 138: The inversion is performed on a Monthly basis (due to the observation gaps of the satellite). Please elaborate in more detail what this means. Does this include that the wind fields are averaged over a month. Or where the daily observations for TROPOMI and the

meteorological data used and the emission assumed to be constant for the one month.

Response: Thank you for your careful review and for raising this concern. According to reviewer 2's suggestion, we have updated our inversion framework to operate on a daily basis in the revised manuscript, instead of the previous monthly configuration. This change improves the temporal resolution of the inverted emissions and allows us to better resolve short-term variations, especially those associated with rapid wartime changes.

To address the issue of limited valid satellite pixels over Ukraine, particularly under persistent cloud cover, we adopted a 7-day moving average of TROPOMI $NO_2$ vertical column densities (VCDs) as observational constraints. This approach balances the need for sufficient observational coverage and the ability to retain temporal signals, as supported by sensitivity tests (see Fig. R3). We also discussed the potential limitation of this smoothing approach in capturing short-lived emission events.

Regarding model treatment, hourly $NO_2$ concentrations from the GEOS-Chem model were sampled at satellite overpass times. No temporal averaging was applied to the wind fields or model meteorology; we used native-resolution meteorological data (from GEOS-FP) to drive the transport and chemistry. The model outputs were vertically regridded using pressure levels to match the vertical resolution of TROPOMI, and we applied the tropospheric averaging kernel and tropopause pressure cut-off following the TROPOMI user manual recommendations.

[Figure]

**Fig. R3.** (a) Number of valid TROPOMI observation grids over Ukraine from 2019 to 2023 under different moving average window lengths. (b) Comparison of daily TROPOMI $NO_2$ VCDs over Ukraine using 7-day and 11-day moving averages. (Fig.S1 in revised manuscript)

We have revised the relevant sections of the manuscript (Lines 164-175):

Furthermore, when using TROPOMI NO$_2$ VCD as observational constraints for inversion, the quality of satellite observations directly impacts the inversion accuracy. In Ukraine, the lack of sufficient valid pixels severely hampers the reliability of daily inversions. We applied a multi-day moving average window to constrain the daily simulated concentrations to address this limitation. The effectiveness of this approach has been demonstrated in previous studies (e.g., Zheng et al., 2020). We evaluated the number of valid TROPOMI observation pixels over Ukraine using different moving average window lengths during the study period (Fig. S1a). When the window size reached 7 days, the data loss significantly decreased, and the proportion of valid grid cells exceeded 90%. However, applying a multi-day average may suppress short-term fluctuations in NO$_2$ concentrations, potentially limiting the ability to capture rapid changes associated with wartime dynamics. We compared the NO$_2$ VCDs over Ukraine using 7-day and 11-day moving averages (Fig. S1b). The results showed no substantial difference between the two, but the 11-day average tended to smooth out peak concentrations more strongly. Based on this assessment, we adopted the 7-day moving average of TROPOMI NO$_2$ VCDs to constrain the model simulations in this study.

And Lines 183-187:

We output hourly NO$_2$ concentrations from the GEOS-Chem model and sampled the values corresponding to the satellite overpass times. Using pressure of each model layer, we vertically regridded the model output to match the vertical resolution of the TROPOMI satellite. Following the recommendations in the TROPOMI User Manual (van Geffen et al., 2019), based on the vertical level of the tropopause provided in the satellite product and the tropospheric averaging kernel, we integrated the tropospheric NO$_2$ profiles to calculate the simulated NO$_2$ VCD for each model grid cell.

3. P6/7 L188: The assumption if the dominant sector is really constant over time is not justified– Also the authors themselves state that it might not be correct. However I am afraid that large parts of the conclusions are based on this assumptions. Or is there any other way to distribute the NO$_x$ emissions among the different sources?

Response: Thank you for this comment. In the revised manuscript, we have updated the method for sectoral allocation of NO$_x$ emissions to improve the robustness of our analysis. We now utilize the sectoral distribution ratios from the updated EDGAR inventory, which extends to 2022. For all years up to and including 2022, the posterior NO$_x$ emissions in each grid cell are distributed across sectors according to the corresponding year's sectoral ratios in the EDGAR inventory.

For 2023, due to the lack of updated sectoral data, we adopted the 2022 sectoral proportions as a proxy. This approach avoids the unrealistic assumption of a temporally invariant sector mix over the entire study period, and instead only assumes relative sectoral stability between 2022 and 2023. We explicitly acknowledge this assumption and its potential limitations in the revised text.

Moreover, our main conclusions regarding total NO$_x$ emission trends and regional anomalies are not solely dependent on the sectoral breakdown. While we do report and discuss sectoral patterns, the key findings related to spatial and temporal variability in emissions are based on total posterior emissions, which are directly constrained by TROPOMI observations.

We have revised Section 2.4 of the manuscript to clearly explain the updated sectoral allocation method (Lines 221-223):

For sectoral emissions, we allocated grid-level totals based on the sectoral distribution ratios provided in the EDGAR inventory. For the year 2023, due to the absence of updated sector-specific data, we assumed that the sectoral proportions in 2022 remained unchanged and applied them to the 2023 total emissions.

4. P13 L339: The prior emissions are stated to be overestimated by 80%. This I s quite large has this been confirmed by similar studies?

Response: Thanks! The 80% overestimation in the original manuscript does not refer to the overestimation of the prior inventory, but rather the prior simulated $NO_2$ VCDs. We compared the CEDS inventory with the inverted 2019 emissions, and the relative deviation between the two was 29.7%.

In the revised manuscript, we have adopted new versions of TROPOMI $NO_2$ VCDs as constraints, used the EDGAR inventory as prior emissions and updated the inversion method. The modeled $NO_2$ VCDs using EDGAR are slightly lower than the TROPOMI v2.4.0 observations overall, but show good agreement during the winter months. Our top-down estimates are approximately 40% higher than the EDGAR inventory, primarily during the spring season. This discrepancy, however, remains within the uncertainty range reported by EDGAR.

We have revised this section in the manuscript, changing the changes in emissions to concentrations (see Lines 412-414).

Prior simulated $NO_2$ VCDs exhibited an underestimation of 11.6% at the national level, with the greatest discrepancy observed in the southwest and northeast regions.

Furthermore, we have added a description of the differences of emissions in prior inventory and inversion in the Uncertainty analysis (see Lines 510-511).

The inversion emissions increased by 39.9% (±7.2%) compared to the prior inventory, which is within the 50% uncertainty range of EDGAR (Crippa et al., 2018).

5. Figure 2: The Crimea peninsular in the South of Ukraine has been occupied since 2014. What causes the $NO_x$ reduction there?

Response: Thank you for this insightful comment. We analyzed the updated results over the Crimea Peninsular to characterize emission changes and their potential drivers. Overall, anthropogenic $NO_x$ emissions increased across the peninsular in both 2022 and 2023. The increase in 2022 was relatively modest, with slight decreases observed in western areas.

This spatial pattern is consistent with Crimea's role during the conflict period. As a key military and logistical base for Russian operations, including hosting air force infrastructure and supply

chains, the region likely experienced intensified activity during the war, contributing to increased $NO_x$ emissions from fuel combustion and transport.

We have added a discussion of this region in the revised manuscript (see Lines 352-360):

In 2022, $NO_x$ emissions in Crimea showed a slight decrease in the western region and a modest increase in the east, primarily due to the concentration of Russian military logistics and air force operations in the eastern part of the peninsula. In 2023, emissions increased further across the region, reflecting Crimea's growing role as a long-term base for Russian troop deployment and logistical support. Sectoral analysis reveals varying degrees of recovery in agriculture, industry, and transportation across central and western Ukraine. In contrast, residential emissions in 2023 remained at levels comparable to those in 2022, consistent with the national trend. These findings suggest that while efforts to reestablish agricultural and industrial activity have taken hold in the rear regions of the battlefield, residential activity has yet to recover, likely due to continued population displacement.

**Technical comments**

1. p3 l 93: the sentence beginning with Meteorological data can be split into two, remove the word "while".

Response: Thank you for the suggestion. We have removed "while" and split that sentence into two sentences in the revised manuscript (see lines 106-108).

2. p5 l141: "global" instead of "g lobal".

Response: Thank you! We have corrected this typo error in the revised manuscript (see Line 177).

---

## Author Comment (AC2)

Dear Reviewer:

We would like to thank the anonymous referee for his/her comprehensive review and valuable suggestions. We have carefully considered each of the referee's comments and suggestions and have revised the manuscript accordingly. In this response, we respond to all of the comments point to point. The referee's comments are listed below in black, our response is given in blue and the modification to the manuscript is listed in red. The page and line numbers for corrections are referred to the revised manuscript; the page and line numbers of the original review manuscript remain unchanged. References relevant to the response are listed at the end of this document.

**Comments:**

In this manuscript Mao et al. evaluate the change in nitrogen oxide emissions in Ukraine due to the Russia-Ukraine war in 2022 using inversion methods. Overall, the discussed merit is interesting an deserves investigation. Before I can recommend publication in ACP, the authors need to address some fundamental aspects.

**Mayor comments**

1. I would suggest to rethink your title. Your methodology relies on inversion techniques and uses TROPOMI data as an input. The phrase "Satellites reveal" is thus misleading. In addition the 28% is associated with uncertainties and I would thus remove it from the title.

Response: We sincerely appreciate the reviewer's suggestion. We have revised the title to "Inversion-based assessment of anthropogenic $NO_x$ emission changes in Ukraine during the 2022–2023 war using TROPOMI satellite data".

2. I would expect a significant contribution of direct and indirect war related $NO_x$ emission (e.g., infrastructure fires). In your study, however, you seem to not include any war related emissions. Keeping in mind the fast changing nature of this war, what uncertainties does this introduce to your methodology and your results? To what degree do you think did war related emissions compensate the reductions reported in your study?

Response: Many thanks for this important and thought-provoking comment. We fully agree that the war itself may emits large amounts of $NO_x$, especially during periods of intense fighting. We have added a new subsection in the Discussion section to discuss the extent to how the war related emissions offset the reductions in anthropogenic $NO_x$ emissions (Fig. R1). Leveraging the spatiotemporally volatile nature of war-related emissions, we compared daily emission estimates during the war with corresponding 30-day Locally Estimated Scatterplot Smoothing (LOESS) emissions. By identifying grid cells with daily anomalies exceeding three standard deviations from the smoothed trend, we delineated "hotspot" regions directly affected by military operations. We then quantified the relative differences between daily and smoothed emissions for both hotspot and non-hotspot areas.

The identified hotspots correspond well with regions experiencing intense military activity, and exhibit significantly higher daily emissions than the smoothed baseline, suggesting the presence of short-term, conflict-induced emission spikes. In contrast, non-hotspot regions showed minimal differences between daily and smoothed values. This analysis suggests that wartime activities offset approximately 8% and 10% of the socio-economics related $NO_x$ emission reductions in 2022 and 2023, respectively.

When these war-related signals were removed, the inversion of 2022 emission reductions exhibited good agreement with the prior EDGAR inventory and independent macroeconomic indicators. This helps to partially explain the lower total reduction than that estimated by bottom-up inventories or energy statistics.

[Figure]

**Fig. R1 Spatiotemporal distribution of significant daily anthropogenic $NO_x$ emission anomalies in Ukraine during the 2022 war period (after 24 February) and 2023.** (a, b) Spatial distribution of daily cumulative significant emission anomalies relative to the LOESS-smoothed emissions in 2022 (a) and 2023 (b). (c, d) Seasonal variation of daily emissions and LOESS-smoothed emissions (c), and corresponding relative differences (d), for 2022 in war-affected hotspots (grid cells with daily anomalies > 3σ) and non-hotspot regions. (e, f) Same as (c, d), but for 2023. (Fig.4 in the revised manuscript)

We have added this part in section 4.3 in the revised manuscript (see lines 469-507):

As mentioned above, War itself may also lead to large amounts of $NO_x$ emissions, including military vehicles and artillery. Thus, we further assessed the war-related $NO_x$ emissions. We firstly applied Locally Estimated Scatterplot Smoothing (LOESS) and residual analysis to quantify the spatial heterogeneity of significant emission anomalies in the 2022 wartime period and 2023 (Fig. 4a, b). Significant emission anomalies were identified by calculating the cumulative significant residuals (exceeded three times the standard deviation) of daily grid-level emissions relative to the corresponding 30-day LOESS-smoothed values. The results indicate localized positive residual anomalies (red

hotspots) were observed along conflict frontlines and logistical hubs, suggesting that military operations and emergency responses significantly elevated emissions in these areas. This is consistent with the severely damaged areas identified by Priyanka Gupta et al. (2024) using NASA MODIS FIRMS active fire detections. We analyzed daily anthropogenic $NO_x$ emissions alongside verified reports of military activity in Ukraine (2022–2023) from BBC and Reuters, and found strong correspondences between emission anomalies and key military events. In Kyiv and Kharkiv, emissions sharply declined following the outbreak of war on 24 February 2022, reaching minima during periods of civilian shutdown. Emissions rebounded during March as military logistics and emergency operations intensified. Kharkiv and Luhansk showed short-term positive anomalies during Ukrainian counteroffensives and Russian reinforcements, while Donetsk experienced sustained negative anomalies due to prolonged conflict and infrastructure damage. Crimea, as a logistical hub, showed persistent emission increases linked to military operations. These findings highlight the potential of high-frequency $NO_x$ emissions as a proxy for monitoring the intensity and evolution of wartime activities.

The emission anomalies observed in the inversion results suggest that wartime activities made a non-negligible contribution to overall $NO_x$ emissions. By comparing daily emissions with their corresponding LOESS-smoothed values, we classified each day's spatial emissions into war-affected hotspots (grid cells with daily anomalies $> 3\sigma$) and non-hotspot regions. Results show that during the war periods of 2022 and 2023, the smoothed emissions in hotspot areas were 30.8% and 35.6% lower than the corresponding daily emissions, respectively. In contrast, differences in non-hotspot regions were only 6.7% and 8.2%. This indicates that smoothing effectively filtered out the high-frequency variability associated with military activities in hotspot areas. With this method, we further estimated the relative deviation of the smoothed emissions from the pre-war baseline, finding reductions of 23% in 2022 and 18% in 2023, which closely matching the emission decreases reported by the EDGAR inventory. This suggests that military-related activities offset approximately 8% and 10% of the overall emission reductions in 2022 and 2023, respectively, which partially explains the lower reductions in our inversion than those from bottom-up inventories and independent economic data. However, quantifying the exact compensatory effect of direct war emissions on emission reductions remains methodologically challenging. Because these sources are inherently episodic and spatially concentrated, complicating their separation from background variability in sectoral inventories. These findings highlight warfare as a distinct emission modulator that can temporarily reshape regional source profiles, though its aggregate contribution remains secondary to economy-wide suppression effects in determining net emission trajectories.

3. Please provide further details on the TROPOMI retrievals. The VCDs are obtained from a polar orbit meaning that the same time of day is observed. How does this impact your methodology, especially considering a shift in activities to the night?

Response: Many thanks for this suggestion. Indeed, TROPOMI aboard the Sentinel-5P satellite is in a sun-synchronous orbit and provides $NO_2$ vertical column densities (VCDs) around local early afternoon (~13:30 LT). As a result, the retrievals reflect only daytime emissions, and can not capture nighttime emission changes. In our inversion framework, we use the TROPOMI $NO_2$ data to constrain daily anthropogenic $NO_x$ emissions through GEOS-Chem simulations that coincide

with the transit times of the corresponding TROPOMI grid. We have added a description in the revised manuscript in Lines 149-151:

TROPOMI is a UV–visible spectrometer aboard the Sentinel-5P satellite in a sun-synchronous polar orbit, crossing the equator at approximately 13:30 local time. We screened GEOS-Chem simulations that overlapped with the transit time of the TROPOMI grid to participate in the inversion.

We acknowledge that in wartime conditions, especially under curfews or infrastructure damage, activity patterns may shift to nighttime (or be suppressed at night for safety). As TROPOMI cannot observe night emissions, this introduces an uncertainty into how representative our estimates are for the true daily total emissions.

We have added a discussion of this limitation and the potential implications for our results in Discussion of the revised manuscript, see Lines 515-524:

Due to the fixed overpass time of TROPOMI at approximately 13:30 local time, the inversion is constrained to reflect emissions around this midday window, limiting its sensitivity to nocturnal emission variations. This temporal sampling introduces a limitation in capturing potential shifts in emission timing, particularly under wartime conditions. During the Russia–Ukraine war, especially in high-risk zones, there may have been a redistribution of human and industrial activities toward nighttime hours due to safety concerns, power rationing, or tactical considerations such as avoiding aerial detection. While such behavioral shifts could potentially increase nocturnal emissions, the current inversion system is unable to capture these changes due to the absence of nighttime satellite data. Thus, while our results provide a robust estimate of daytime anthropogenic $NO_x$ emissions, they may overestimate total emission changes if substantial nocturnal activity occurred. Future work integrating ground-based measurements, high-temporal-resolution modeling, or geostationary satellite data (e.g., TEMPO, Sentinel-4) could help address this temporal limitation.

4. To improve data quality, you perform the inversion of the anthropogenic $NO_x$ emissions on a monthly scale. Since you frequently highlight the fast changing nature of the war, it sounds that using monthly averaged data is not an optimal choice. To what degree does this influence the predicted results?

Response: Many thanks for this comment. We fully agree that using monthly averaged data may limit the ability to capture short-term changes in emissions, especially under rapidly evolving conditions such as wartime disruptions. In light of your suggestion, we have revised the inversion process and the prior inventory. In brief, in the updated inversion method, we adopt the EDGAR inventory updated to 2022 as the prior, and estimate daily anthropogenic $NO_x$ emissions at a spatial resolution of 0.25°×0.3125° using a 7-day moving average of daily TROPOMI observations. A more accurate prior inventory increases the reliability of the inversion results, and the higher spatiotemporal resolution of the emissions provides strong support for evaluating the impact of rapidly changing war events on anthropogenic emissions.

We have added the following descriptions in the revised manuscript (see lines 164–175):

Furthermore, when using TROPOMI NO$_2$ VCD as observational constraints for inversion, the quality of satellite observations directly impacts the inversion accuracy. In Ukraine, the lack of sufficient valid pixels severely hampers the reliability of daily inversions. We applied a multi-day moving average window to constrain the daily simulated concentrations to address this limitation. The effectiveness of this approach has been demonstrated in previous studies (e.g., Zheng et al., 2020). We evaluated the number of valid TROPOMI observation pixels over Ukraine using different moving average window lengths during the study period (Fig. S1a). When the window size reached 7 days, the data loss significantly decreased, and the proportion of valid grid cells exceeded 90%. However, applying a multi-day average may suppress short-term fluctuations in NO$_2$ concentrations, potentially limiting the ability to capture rapid changes associated with wartime dynamics. We compared the NO$_2$ VCDs over Ukraine using 7-day and 11-day moving averages (Fig. S1b). The results showed no substantial difference between the two, but the 11-day average tended to smooth out peak concentrations more strongly. Based on this assessment, we adopted the 7-day moving average of TROPOMI NO$_2$ VCDs to constrain the model simulations in this study.

[Figure]

**Fig. R2.** (a) Number of valid TROPOMI observation grids over Ukraine from 2019 to 2023 under different moving average window lengths. (b) Comparison of daily TROPOMI NO$_2$ VCDs over Ukraine using 7-day and 11-day moving averages. (Fig.S1 in the revised manuscript)

Based on the determination that a 7-day moving window sufficiently meets the requirements for the inversion, we have updated the inversion methodology section accordingly in the revised manuscript (see Lines 203–206):

Due to the reduced TROPOMI observation coverage on specific days, as discussed in Sect. 2.3, we

employed 7-day moving averages of satellite $NO_2$ observations for comparison with daily simulated VCDs in this study. Specifically, for each day during the study period, the simulated $NO_2$ VCDs were constrained using the quality-filtered TROPOMI $NO_2$ VCD observations. This approach has been demonstrated to be effective by Zheng et al (2020).

5. I was surprised to see such a coarse model resolution being used when focusing on such a small region. At the same time, TROPOMI provides data at a km scale. How can you justify such a coarse resolution knowing that other inverse modelling infrastructures provide resolutions at the km scale? How does this affect your results?

Response: Thank you for this thoughtful comment. We agree that the spatial resolution of the inversion is an important aspect, especially when working with high-resolution satellite data like TROPOMI. In our initial submission, we used monthly averaged data and a relatively coarse model resolution. However, in response to your above comments, we have revised our inversion strategy, and now use daily TROPOMI data to perform the inversions at $0.25° \times 0.3125°$ resolution. These revisions have already been detailed in our above responses (see the response to comment on data averaging and method changes). Although the resolution of $0.25° \times 0.3125°$ is still relatively low, it represents the highest regional resolution supported by GEOS-Chem Classic, which we used in this study.

We also explicitly discuss this limitation and its implications in the revised manuscript in Lines 536-538:

In addition, the inversion was conducted at a spatial resolution of $0.25° \times 0.3125°$, which may smooth or omit localized emission signals, leading to potential biases in the estimated $NO_x$ emissions.

**Minor comments**

1. Line 13: Replace "economic production" with e.g. "society".

Response: Thanks! In the revised manuscript, we have replaced "economic production" with "society" in Line 13 as recommended. The revised text now reads: "The outbreak of the Russia–Ukraine war in 2022 brought a huge impact on the Ukrainian society." (Line 13)

2. Line 18: Your abstract only mentions decreases in $NO_x$ emissions, even though you document increased emissions in urban areas in West Ukraine. This can be misleading and should be mentioned in the abstract.

Response: Many thanks for this suggestion. We have revised the abstract to explicitly mention both the overall decreasing trend and the localized increases in urban areas of West Ukraine. The updated text now reads (see Lines 18-21):

Regionally, Eastern Ukraine experienced larger reductions in $NO_x$ emissions in both 2022 and 2023 by 29% and 17%, respectively, due to direct damage from frontline military operations. In contrast,

Western Ukraine experienced a relatively modest emission reductions of only 8% in 2022 with emissions increasing in some regions. In 2023, the emissions increased in most western regions.

3. The introduction would greatly benefit from a figure which shows yearly average $NO_x$ VCD from TROPOPMI for the base year as well as 2022.

Response: Many thanks for this suggestion. We have added the spatial distribution of the relative changes in TROPOMI $NO_2$ vertical column densities (VCDs) during the wartime period in 2022 and throughout the year 2023, compared to the corresponding baseline periods in the Satellite $NO_2$ VCD Observations subsection in the revised manuscript. This figure provides a visual overview of spatial changes in $NO_2$ concentrations over Ukraine and helps set the stage for the inversion-based emission analysis. The corresponding description has been included in the revised manuscript to enhance clarity and context (Line 156-160):

In this study, we quantified relative changes in TROPOMI $NO_2$ VCDs between the wartime period in 2022 and full year of 2023 and the corresponding periods in pre-war baseline (mean of 2019 and 2021; excluding 2020 due to COVID-19 anomalies, same thereafter) (Fig. 1). Results demonstrate that satellite $NO_2$ VCDs effectively capture spatiotemporal variability of air pollution during the war, though their representation of ground-level emissions remains limited given their tropospheric column nature.

[Figure]

**Fig. R3 The Spatial changes in satellite-observed NO₂ VCDs are illustrated (a) during the war (March to December) in 2022 and (b) in 2023 relative to the corresponding periods of baseline (average of 2019 and 2021).** (Fig.1 in the revised manuscript)

4. Line 196: Please elaborate on the factors 1.2 and 0.7.

Response: Thank you for your comment. We agree that the originally used perturbation factors (1.2 and 0.7) in the OSSE lacked a clear justification and could be misleading. In the revised manuscript, we have updated this section to provide a more scientifically grounded basis for the assumed emission perturbations. Specifically, we now assume that the true emissions in 2022 are 0.5 times the EDGAR prior, based on the reported average uncertainty of $NO_x$ emissions over the European Union (~51.7%) in the EDGAR inventory (Crippa et al., 2018). For 2023, we assume the true emissions are 1.25 times those of 2022, guided by the observed interannual variability in TROPOMI NO₂ VCDs, which exceeded 25% in some months.

These changes are now reflected in the revised manuscript (Line 232-237), and we believe they provide a more realistic and traceable basis for evaluating the OSSE performance:

In the OSSE, we used the EDGAR emissions in 2022 as the prior, assuming that the true emissions in

2022 were 0.5 times the prior, and the true emissions in 2023 were 1.25 times the true values in 2022. The assumed true emissions for 2022 were informed by the average uncertainty of $NO_x$ emissions over the European Union as reported by the EDGAR inventory, which is approximately 51.7% (Crippa et al., 2018). The assumed true emissions for 2023 were guided by the interannual variability of TROPOMI $NO_2$ VCDs, which showed that monthly $NO_2$ concentrations in 2023 differed from those in 2022 by more than 25% in certain months.

5. Line 244: How do you account for the population migration in your emission datasets?

Response: Thank you! We fully agree that population migration could influence anthropogenic $NO_x$ emissions, especially in the context of large-scale displacement during the war. However, we were unable to obtain spatially and temporally resolved population migration data with sufficient accuracy to be integrated into our emission estimates or inversion framework.

As per your suggestion, we have revised the manuscript to emphasize the lack of quantitative data on population displacement. The updated sentence reads (Lines 281-286):

After September, the $NO_x$ emissions showed a new round of decline of 23.8% (±4.3%), reaching a peak decline of 30.5% (±5.5%) in December. This was primarily due to the increased energy demand in the baseline years and intensified energy shortages in 2022 during the cold season. Notably, although we cannot precisely quantify the contribution of population displacement to emission reductions, the continued outflow of residents due to the ongoing impacts of the war likely contributed to the enhanced decrease in wintertime $NO_x$ emissions.

6. Line 271: What "drivers" are you referring to? How does military activity related to transport compensate these changes?

Response: Thank you for your helpful question. We realize that the term "drivers" in the original sentence was ambiguous. To improve clarity, we have revised this part of the manuscript to explicitly discuss the sector-specific seasonal patterns during the war, based on a detailed comparison of anthropogenic $NO_x$ emissions from each sector between the war years and the baseline period. In the revised manuscript, we removed the vague reference to "drivers" and instead provided a quantitative analysis showing that during the early months of the war (March–May 2022), the transport sector initially experienced a sharp decline in emissions due to conflict-related disruption, including damaged infrastructure and reduced civilian mobility. However, the reduction was less pronounced than in other sectors, and even partially rebounded in subsequent months, possibly due to military transport activities, humanitarian logistics, and evacuation-related movement. For example, transport-related emissions declined by 24% (±4.3%) in March 2022 and only slightly further in April. These updates help clarify that while infrastructure damage led to a persistent reduction in transport emissions, increased demand from military-related mobility may have partially compensated for what would otherwise have been a deeper decline.

Please see the Lines 325-328 in the revised manuscript:

The transportation sector demonstrated notable responsiveness to increased transportation demand during the war, with a 24% (±4.3%) decline in emissions observed in March, followed by a slight rebound. Owing to military transport activities, humanitarian logistics, and evacuation-related movement, the transport emissions was smaller than others in 8% (±1.4%).

7. Line 417: Please elaborate on what policies you are referring to.

Response: Thank you for the suggestion. We have revised the sentence in the conclusion to clarify the types of policies our findings may inform. Specifically, we now refer to post-war reconstruction planning, energy security strategies, and emission mitigation policies, which are all areas where understanding wartime emission dynamics and infrastructure impacts can provide valuable insights (see the Lines 558-560):

While our analysis is only a snapshot of the impacts of war, the findings have far-reaching implications for further research and for informing post-war reconstruction planning, energy security strategies, and emission mitigation policies.

8. Fig. 1 and 2: Please fix the inconsistencies in the x-axis labels.

Response: Thank you for pointing this out. We have carefully reviewed and corrected the inconsistencies in the x-axis labels of all Figures in the revised manuscript to ensure uniform formatting and clarity. The updated figures now use consistent date formats and labeling intervals across both panels. We believe this improves the visual coherence and readability of the figures.

---

## Author Response (AR2)

Dear editor.

We would like to thank the reviewer for his/her constructive and detailed comments,

which have helped us to substantially improve our manuscript. According to the

reviewer's comments and suggestions, we have made revision to our manuscript, the

main modifications are as follows:

1. Clarification of TROPOMI NO2 product versions. We have added a detailed

description of the differences between the v2.4.0, v2.5.0, and v2.6.0 datasets, and

discussed the potential impact on our inversion results.

2. Several other revisions were made to enhance the manuscript's clarity and

consistency, including:

1) We supplemented the introduction with background information on the changes in

anthropogenic NOx emissions in Ukraine before the Russia–Ukraine war.

2) All abbreviations are now defined upon first use to avoid ambiguity.

3) References have been added for all datasets used in the study.

4) Wording throughout the manuscript has been refined, and certain sentences have

been reorganized to improve logical flow.

5) A disclaimer has been included to emphasize the authors' neutral stance regarding

the war and to note that some maps may contain disputed territories.

We believe that these revisions fully address the reviewer's concerns and strengthen the

manuscript's overall contribution. For your convenience, we have submitted both a

clean revised manuscript and a marked-up version showing the changes, along with a

point-by-point response to the reviewers' comments.

We once again thank you and the reviewers for the valuable feedback and guidance. We

look forward to your evaluation for our revised submission and hope that it will meet

the standards of Atmospheric Chemistry and Physics.

Best regards,

Professor Fei Jiang

Nanjing University

E-mail: jiangf@nju.edu.cn

**Dear Reviewer:**

We would like to thank the anonymous referee for his/her comprehensive review and valuable suggestions. We have carefully considered each of the comments and suggestions and have revised the manuscript accordingly. In this response, we respond to all of the comments point to point. The referee's comments are listed below in black, our response is given in blue, and the modification to the manuscript is listed in red. The page and line numbers for corrections are referred to the revised manuscript; the page and line numbers of the original review manuscript remain unchanged. References relevant to the response are listed at the end of this document.

**Comments:**

The revised version of the manuscript by Mao et al. substantially improved, and most of my comments have been satisfactorily addressed. However, before I can recommend publication in ACP, the authors need to address the following aspects:

**Major comment:**

In the revised methodology, different versions of the TROPOMI NO2 VCD data are used for different time periods. Rather than using v2.3.1, you use versions v2.4.0, v2.5.0 and v2.6.0 for different time periods. You also highlight that these versions differ substantially from each other, with improvements in the later versions compared to the previous ones. Why don't you use the same version for the entire time period? What uncertainty does this introduce to your methodology? Please provide an overview of the differences between the versions in the SI and include a discussion of the introduced uncertainties in section 4.4.

Response: Thank you for raising this important issue. The reason for using different versions of the TROPOMI NO2 dataset for different periods is that there is no single version covered the entire study period (2019–2023). For instance, version v2.4.0 only updated to 12 March 2023, while version v2.5.0 only covered the period from 12 March to 26 November 2023 (see https://www.temis.nl/airpollution/no2col/tropomi\_no2\_data\_versions.php). According to the Sentinel-5P TROPOMI NO2 ATBD (Van Geffen et al., 2024), the differences between these versions (v2.4.0, v2.5.0 and v2.6.0) are relatively minor compared with the substantial changes that occurred between v2.3.1 and v2.4.0.

Compared to v2.3.1, the algorithm in v2.4.0 has introduced an improved treatment of the air mass factor (AMF) calculation, updates in the absorption cross-sections, and improved handling of surface albedo and cloud parameters. These changes led to a better consistency between satellite retrievals and independent validation datasets.

For v2.5.0, the algorithm implemented a correction in the qa\_value flagging, especially for snow/ice conditions, thereby slightly increasing the number of valid observations without altering the underlying NO2 retrieval algorithm.

The algorithm of v2.6.0 incorporated an update in the FRESCO cloud algorithm, improving cloud pressure retrievals used in the AMF calculation. This change may cause small shifts in the tropospheric NO2 VCD, but validation shows the effect is within a few percent and largely regional.

The uncertainties introduced by these version transitions are therefore limited. The most relevant difference arises between v2.6.0 and the earlier versions because of the cloud-related update, but in our study this version contributes only one month of data (December 2023). Its influence on the overall results is thus negligible.

Following the reviewer's suggestion, we have 1) added explanations for using different data versions, 2) added some discussion for its potential impact on the results, and 3) added a concise overview of these version changes in the Supplementary Information (Table. R1).

**1) Explanations for using different data versions (see Lines 146~150):**

Different versions of the dataset were used for different time periods because each version only covers a specific time period (see https://www.temis.nl/airpollution/no2col/tropomi\_no2\_data\_versions.php). These products incorporate improved and consistent Level-1b processing and retrieval algorithms, with only minor adjustments between versions (Table S1), representing the most up-to-date and accurate TROPOMI NO2 dataset available.

**2) added some discussion for its potential impact on the results (see Lines 516~521):**

In addition, the use of different product versions (i.e., v2.4.0, v2.5.0, and v2.6.0) across the study period may introduce further uncertainties. While these versions are largely consistent with each other and share the same retrieval algorithm framework, minor differences exist due to bug fixes and updates in quality assurance flagging in v2.5.0 and improvements in cloud pressure retrievals affecting air-mass factor calculations in v2.6.0 (Table S1). These differences may lead to small regional or temporal shifts in retrieved NO2 VCDs, potentially propagating into the inversion results.

3) Added a concise overview of these version changes in the Supplementary Information:

Table. R1. Key updates and their potential impact on NO2 VCD of versions v2.4.0~v2.6.0 of TROPOMI NO2 VCDs (Table. S1 in the revised SI)

| Version | Time period                   | Key updates                                                                                       | Potential impact on NO 2 VCDs                                                   |
|---------|-------------------------------|---------------------------------------------------------------------------------------------------|--------------------------------------------------------------------------------------------|
| v2.4.0  | 1 May 2018 – 12
Mar. 2023  | Improved Air-mass factor (AMF) (surface albedo, clouds), updated cross-sections                   | Better consistency with validation;
no major discontinuity                              |
| v2.5.0  | 13 Mar. – 26 Nov.
2023     | Bug fix in quality assurance value (qa_value) (snow/ice handling)                                 | Slightly more valid pixels;
minimal effect on mean NO 2                      |
| v2.6.0  | 26 Nov. 2023 – 8
Sep. 2024 | Fast Retrieval Scheme for Clouds
from the Oxygen A band cloud
pressure update affecting AMF | Small regional shifts in VCD (few percent);
limited in our study due to short time span |

**Minor comments:**

1. In your manuscript, you focus primarily on relative changes. The introduction would be improved by providing some context on absolute  $NO_x$  emissions in Ukraine. Additionally, it would be helpful if you could provide some background information on whether Ukraine was reducing its emissions prior to 2019.

Response: Thank you for this constructive suggestion. We have revised the introduction to include the changes in  $NO_x$  emissions in  $2015\sim2019$  based on the EDGAR v8.1 inventory. From 2015 to 2019, Ukraine's anthropogenic  $NO_x$  emissions were in the range of 504 to 541 kt/yr, indicating that the  $NO_x$  emissions were rather stable. This revised description has been added to the introduction (Lines  $61\sim63$ ).

According to the Emissions Database for Global Atmospheric Research (EDGAR) v8.1 (Crippa et al., 2024), the annual  $NO_x$  emissions in Ukraine were rather stable before the Russia–Ukraine war, with relative changes in the range of -5% to 3% from 2015 to 2019.

2. Line 58-60: What about particle formation?

Response: Thank you for pointing this out. We have revised the text accordingly, see Lines 57~60 in the revised manuscript and as follows:

As a short-lived gas, directly emitted nitric oxide (NO) can be rapidly oxidized to form nitrogen dioxide (NO2), which, in the presence of sunlight and volatile organic compounds (VOCs), contributes to net ozone (O3) generation and secondary particulate matter formation (Roger Atkinson, 2000).

3. Line 107: Please also provide the approximate grid size in km for Ukraine.

Response: Thank you! We have provided the approximate grid size in kilometers. The model resolution of  $0.25^{\circ} \times 0.3125^{\circ}$  corresponds to about  $14.0 \sim 20.7$  km  $\times$  34.8 km over Ukraine (Lines  $108 \sim 110$ ):

The model was operated at a horizontal resolution of  $0.25^{\circ}$  (latitude)  $\times$   $0.3125^{\circ}$  (longitude), corresponding to approximately 17 km  $\times$  35 km over Ukraine, and a vertical resolution of 47 layers.

4. Line 343: Change to "The war has had and continues to have direct and indirect impacts on [...]"

Response: Thank you! We have revised the sentence to (Lines 350~351):

The war has had and continues to have direct and indirect impacts on industrial production across Ukraine, with the most pronounced damage observed in conflict zones.

5. Line 423: What does "SSCU" stand for? Please introduce all abbreviations properly and provide a reference for the data source.

Response: Thank you for pointing this out. We have clarified the abbreviation in the manuscript. There is a typo here, SSCU should be SSSU, which refers to the State Statistics Service of Ukraine. In addition, we have checked abbreviations throughout the manuscript and added the full name and reference before all the first abbreviation (Lines 434~436).

Further analysis using the data from State Statistics Service of Ukraine (SSSU, 2025) reveals that oil and natural gas consumption in Ukraine decreased by 15% and 34% in 2022, and by 13% and 32% in 2023, respectively (Table 1).

6. Table 1: Please provide a reference for the "SSCU statistics".

Response: Thank you. We have now added the reference for the "SSSU statistics" in Table 1. Specifically, the data are from the State Statistics Service of Ukraine (SSSU), and the official website has been cited in the reference list.

7. Some of the wording needs to be more scientific. For example, the word "fell" (line 22) is ambiguous when describing emission reductions. Please review your manuscript accordingly.

Response: Thank you for this helpful suggestion. We agree that the wording should be more scientific. We have carefully checked the manuscript and replaced ambiguous expressions such as "fell" with more precise terms (e.g., "decreased", "declined", or "was reduced") when referring to changes in emissions. We have also conducted a

thorough review of the entire text, correcting certain verb tenses and sentence structures.

**References**

Van Geffen, J. H. G. M., Eskes, H. J., Boersma, K. F., and Veefkind, J. P.: TROPOMI ATBD of the total and tropospheric NO2 data products document number: S5P-KNMI-L2-0005-RP, KNMI, 2024.